GROM-RD: resolving genomic biases to improve read depth detection of copy number variants

Smith Sean D.
Kawash Joseph K.
Grigoriev Andrey andrey.grigoriev@rutgers.edu
Department of Biology, Center for Computational and Integrative Biology, Rutgers University , Camden, NJ , USA
Lazo Gerard
Electronic publication date: 2015 Mar 17
Publication date: 2015
Volume: 3
Electronic Location ID: e836
Received 2014 Dec 5; Accepted 2015 Feb 23
Copyright: © 2015 Smith et al.
Copyright year: 2015
Copyright holder: Smith et al.
License: This is an open access article distributed under the terms of the Creative Commons Attribution License, which permits unrestricted use, distribution, reproduction and adaptation in any medium and for any purpose provided that it is properly attributed. For attribution, the original author(s), title, publication source (PeerJ) and either DOI or URL of the article must be cited.
License URL: https://creativecommons.org/licenses/by/4.0/

Keywords: Copy number variant, Genomic bias, Next gen sequencing

Funding: This work was in part supported by the NSF grant DBI-1126052 to AG. The funders had no role in study design, data collection and analysis, decision to publish, or preparation of the manuscript.

==============================
Amplifications or deletions of genome segments, known as copy number variants (CNVs), have been associated with many diseases. Read depth analysis of next-generation sequencing (NGS) is an essential method of detecting CNVs. However, genome read coverage is frequently distorted by various biases of NGS platforms, which reduce predictive capabilities of existing approaches. Additionally, the use of read depth tools has been somewhat hindered by imprecise breakpoint identification. We developed GROM-RD, an algorithm that analyzes multiple biases in read coverage to detect CNVs in NGS data. We found non-uniform variance across distinct GC regions after using existing GC bias correction methods and developed a novel approach to normalize such variance. Although complex and repetitive genome segments complicate CNV detection, GROM-RD adjusts for repeat bias and uses a two-pipeline masking approach to detect CNVs in complex and repetitive segments while improving sensitivity in less complicated regions. To overcome a typical weakness of RD methods, GROM-RD employs a CNV search using size-varying overlapping windows to improve breakpoint resolution. We compared our method to two widely used programs based on read depth methods, CNVnator and RDXplorer, and observed improved CNV detection and breakpoint accuracy for GROM-RD. GROM-RD is available at http://grigoriev.rutgers.edu/software/.

Introduction

Copy number variants (CNVs) have been linked to several diseases including cancer (Berger et al., 2011; Campbell et al., 2010; Stephens et al., 2009), schizophrenia (Stefansson et al., 2009), and autism (Marshall et al., 2008). Compared to single nucleotide polymorphisms (SNPs), structural variants (or SVs, which include CNVs, insertions, inversions, and translocations) account for more differences between human genomes (Baker, 2012) in terms of the number of nucleotides and potentially have a greater impact on phenotypic variation (Korbel et al., 2007). Modern sequencing technologies, often identified as next-generation sequencing (NGS), have enabled higher resolution of CNVs compared to older methods such as array comparative genome hybridization (aCGH) and fosmid paired-end sequencing (Korbel et al., 2007). NGS produces sequenced reads, either single- or paired-end, that are mapped to a reference genome. Several strategies have been developed to detect SVs. Paired-read (PR) methods search for clusters of discordant (aberrant insert size or orientation) read pairs. Split-read methods map previously unmapped reads by splitting the reads. Read depth (RD) methods identify CNVs by detecting regions of low or high read coverage. De novo methods assemble reads into contigs, particularly useful for detecting insertions. Each detection strategy has advantages and disadvantages, and they complement each other by detecting SVs not found or not detectable using the other strategies. For example, RD does not depend on paired reads for finding SVs and is able to detect CNVs with mutated or rough breakpoints that may not be detectable with paired or split reads, but RD is unable to detect insertions, translocations, and inversions.

Several whole genome sequencing (WGS) RD methods, CNV-seq (Xie & Tammi, 2009), SegSeq (Chiang et al., 2009), rSW-seq (Kim et al., 2010), CNAseg (Ivakhno et al., 2010), and CNAnorm (Gusnanto et al., 2012), require a control sample. Other WGS RD methods, such as JointSLM (Magi et al., 2011) and cn.MOPS (Klambauer et al., 2012), require multiple samples. Often multiple samples or a suitable control are not available. Whole exome sequencing (WES) RD methods, including ExomeCNV (Sathirapongsasuti et al., 2011), CONTRA (Li et al., 2012), EXCAVATOR (Magi et al., 2013), CoNIFER (Krumm et al., 2012), and XHMM (Fromer et al., 2012) are limited to detection in coding regions of the genome (Sims et al., 2014). WGS RD methods that do not require a control include FREEC (Boeva et al., 2011), ReadDepth (Miller et al., 2011), CNVnator (Abyzov et al., 2011), and RDXplorer (Yoon et al., 2009).

Detecting CNVs is complicated by GC bias of NGS technologies, whereby read coverage varies depending on the GC content of the genome region. Existing RD methods reduce GC bias by GC bin mean normalization (CNVnator and RDXplorer), polynomial fitting (FREEC), and LOESS regression (ReadDepth). However, these methods do not consider differences in read depth variance with GC content, which may exist after GC bias correction. Complex and repetitive regions are challenging for all CNV detection methods including RD. Complex regions near telomeres and centromeres are known to be SV hotspots (Mills et al., 2011) and sequencing bias has been observed in repeat regions (Ross et al., 2013). However, RD methods have not been tailored for the difficulties of complex and repetitive regions. Additionally, RD methods suffer from low breakpoint resolution.

We have developed GROM-RD, a control-free WGS RD algorithm with several improvements and novel features compared to existing RD algorithms, such as excessive coverage masking, GC bias mean and variance normalization, GC weighting, dinucleotide repeat bias detection and adjustment, and a size-varying sliding window CNV search. These features address weaknesses in existing RD methods and biases in genomic sequencing that limit CNV sensitivity, specificity, and breakpoint accuracy, as evidenced by comparison of our algorithm to two most commonly used control-free WGS RD tools, RDXplorer (Yoon et al., 2009) and CNVnator (Abyzov et al., 2011). GROM-RD showed improved predictive capabilities and breakpoint resolution for CNVs, as well as excellent scalability for different NGS datasets, both simulated and real.

Methods

GROM-RD outputs a union set from two pipelines that differ based on the inclusion or exclusion of a pre-filtering step, excessive coverage masking (Fig. 1). Each step from Fig. 1 will be described in the following subsections.

Figure 1 GROM-RD pipeline summary.

Two iterations of the pipeline are combined into a union set of CNV predictions. For the first iteration (step 1 included), CNV detection in stable regions is improved by masking regions of excessive coverage. Without masking (step 1 excluded), CNVs are detected in complex and repetitive regions that are characterized by excessive coverage.

Excessive coverage masking

Abnormal read coverage has been reported in centromere and telomere regions (Rausch et al., 2012). Similarly, we observed excessive read coverage in certain regions, particularly near centromeres (data not shown). This might be due to complex and repetitive segments, which are common in the human genome and can complicate CNV detection. Such high read coverage may result in false positives and also reduce CNV sensitivity in less complex regions. GROM-RD uses a two-pipeline approach to detect CNVs in complex and repetitive segments and improve sensitivity in less complicated regions. In the first pipeline, we mask clusters of blocks (10,000 base segments) with high read coverage (default: >2× chromosome average) and run GROM-RD on the masked genome. A cluster is defined as a section of the genome where >25% of the blocks have high read coverage and a minimum of four blocks have high read coverage. High coverage regions have been shown to have a high concentration of SVs (Mills et al., 2011). Thus, in the second pipeline, we run GROM-RD on the unmasked genome. GROM-RD outputs a union set of predicted CNVs from the two pipelines. Many false positives may be produced from spikes in read coverage, particularly for the unmasked genome. Thus during later steps in the pipeline, read coverage greater than twice the chromosome average is adjusted (described in ‘GC bias normalization’).

GC weighting

Variation in the GC content of genome regions affects read coverage produced by NGS platforms. A post-sequencing approach used by many RD algorithms, such as CNVnator and RDXplorer, is to bin genome regions by GC content and adjust the average read depth of each bin to the average read depth of the genome, referred to as GC bias normalization. Here we discuss the first step of this approach, calculating GC content of genome regions. RD algorithms often divide a chromosome into regions, referred to as windows, of a fixed size and estimate read depth in each window by counting reads within the window. GC content for a window is calculated from the proportion of reference sequence G and C bases within the window. Previous studies (Aird et al., 2011; Benjamini & Speed, 2012; Bentley et al., 2008) have identified PCR bias as the main contributor to GC bias in NGS. Thus, reference bases outside a window may affect read coverage within a window, especially for long reads and paired-end reads. Benjamini & Speed (2012) showed a higher correlation between GC content and read depth when considering the GC content of the entire PCR-replicated DNA fragment rather than the sequenced segment. Based on these observations, we developed a novel GC weighting method to consider all bases within an average insert size. To maximize sensitivity, we do not calculate GC weighting for a window of bases; instead, GC weighting is calculated for each base i as hi = ∑wjaj/∑wj, where j is a base that may affect read depth for base i, wj is the weight of base j and is equivalent to the sum of average inserts with unique starting locations and that overlap base j and base i, and aj is 1 if base j is a G or C and 0 otherwise. For single-end reads, the insert size is equivalent to read length.

GC bias normalization

As referred to previously, “GC bias” in this context denotes variation in read coverage produced by NGS platforms as a result of variation in the GC content of genome regions. Many RD algorithms, such as CNVnator and RDXplorer, bin genome regions (windows) by GC content and adjust the average read depth of each bin to the average read depth of the genome: (1) ri,norm=rim/mGC

where ri,norm is the read coverage of a window after normalization, ri is the read coverage of window i prior to normalization, m is the global mean read coverage of all windows in the genome, and mGC is the mean read coverage of all windows with similar GC content (Yoon et al., 2009). Although this method normalizes the read depth means across the GC bins, we found differences in variance after GC bias correction (Fig. 2). From this observation, we expect methods using this approach to over-predict CNVs when a GC region has high variance and under-predict CNVs when a GC region has low variance.

Figure 2 Standard deviation after GC bias normalization.

Data produced from chromosome 19 of NA12878 (Illumina high coverage paired-end read dataset aligned with BWA to human reference hg18) (DePristo et al., 2011) using 100-base non-overlapping windows. Reads were assigned to a window if the read center was within the window. After correcting for GC bias using a common approach, the standard deviation varies with GC content. This negatively impacts further analysis by CNV detection algorithms.

We use a quantile normalization approach to correct for variance across bins of GC weighted bases (Lin et al., 2004). For this approach, we rank bases in each bin based on read depth and calculate a rank proportion pi for each base i using (2) pi=Ri/nif 2Ri≤npi=n−Ri/nif 2Ri>n

where Ri is the read depth rank for base i and n is a count of bases with a particular GC weighting. When Ri is 0 (for 2Ri ≤ n) or n − Ri is 0 (for 2Ri > n), the numerator in Eq. (2) is set to 0.5. Subsequently, pi is converted to standard deviation units, xi, using a pre-computed normal distribution table. Note when n is identical for all GC bins and there are no read depth ties within a GC bin, each bin distribution will have identical statistical properties, including mean and variance, after quantile normalization. Statistical properties of quantile normalized distributions may vary across GC bins when n varies, however this effect is negligible when n is large. GROM-RD requires a GC bin to have at least 100 bases. GROM-RD does not produce a normalized read depth as in Eq. (1) because it is not necessary for further analysis. Instead, read depth in standard deviation units is used. As mentioned previously in ‘Excessive coverage masking,’ to reduce false positives, read coverage greater than twice the chromosome average is adjusted by averaging the rank of the observed read coverage and the rank of read coverage equivalent to twice the chromosome average read coverage. CNVs may occur in low mapping quality regions; however, read coverage distributions tend to differ between low mapping quality and high mapping quality regions. To compensate for variation of read coverage distributions with mapping quality, GROM-RD calculates the average mapping quality for each window and creates separate distributions for low mapping quality (default: <5) and high mapping quality windows. The nature of the read depth distribution for NGS data has not been clearly defined. A rank-based approach does not assume a specific distribution and is less affected by outliers when compared to parametric methods.

Dinucleotide repeat bias normalization

Repeat bias has been observed with NGS technologies (Ross et al., 2013). We found similar repeat biases in our investigations. Additionally, these biases may vary with sequencing technology and genomes. For instance, we observed decreased coverage for AT repeats in human (Fig. 3) but not for other genomes (data not shown). We found that dinucleotide repeats as short as 20 bases affected coverage. GROM-RD detects dinucleotide repeat biases and uses a quantile normalization method in the respective genomic regions. Dinucleotide repeats with average read coverage that is more than 1.5 standard deviations below the genome average read coverage, and vice versa (genome coverage more than 1.5 standard deviations above dinucleotide coverage), are considered biased. For a biased dinucleotide repeat, we use a quantile normalization approach similar to our GC bias normalization, except Ri is the read depth rank of occurrence i of a particular dinucleotide repeat. From this we obtain read depth in standard deviation units for each biased dinucleotide repeat occurrence. As we move further from a repeat, GROM-RD creates separate sample distributions in 10 base increments to adjust for the decreasing influence of repeat bias. Thus, we bin bases by distance from the repeat, in contrast to binning by GC weighting as described in ‘GC weighting.’ Repeat bias normalization is applied within a distance of half-insert size from biased dinucleotide repeats. For genomic regions with dinucleote repeat bias, dinucleotide repeat bias normalization replaces GC bias normalization. To our knowledge, GROM-RD is the first RD method to specifically adjust for repeat bias.

Figure 3 Example of dinucleotide repeat bias in a human genome.

AT repeats had lower coverage compared to other dinucleotide repeats for human genome NA12878 (Illumina high-coverage paired-end read dataset aligned with BWA to human reference hg18) (DePristo et al., 2011). Dinucleotide repeats less than 20 bases were filtered. Dinucleotide combinations with less than 50 occurrences in the genome are not shown.

Sliding window CNV search

RD methods typically suffer from reduced breakpoint resolution compared to other methods, such as split-read. One reason for low resolution is fixed-size, non-overlapping windows. We employ sliding windows that sequentially increase in one-base increments to improve breakpoint resolution. Fixed-size, non-overlapping windows also reduce sensitivity when CNVs start or end near the center of a non-overlapping window. Using sliding windows, GROM-RD is equally sensitive to CNVs regardless of start or end points. Additionally, by creating distributions for incremental window sizes, GROM-RD improves sensitivity on a range of CNV sizes.

As described in the previous sections, GROM-RD normalizes GC bias or, if necessary, dinucleotide repeat bias for each base. However, we do not expect to find one base deletions or duplications; instead, GROM-RD combines normalized bases into windows by averaging standard deviation units of all bases in a window. Since the means and variances of the bases have been normalized with respect to GC bias or dinucleotide repeat bias, GC and dinucleotide bias are not associated with the windows.

For each window size, we sample a set of windows from the dataset and obtain a read depth mean and standard deviation. Then, we identify base positions with abnormal read coverage ≥1.3rave,h for duplications or ≤0.70rave,h for deletions (for diploids) as potential breakpoints, where rave,h is the average read depth for bases with h weighted GC content. If at least half of the bases have abnormal coverage for a minimum window size, wl,min (default = 100) beginning at a potential breakpoint j, we calculate a z-score, z, based on a sample distribution of read depths for wl,min and the read depth of a window i having size wl,min and beginning at j.

Several parameters affect calling CNVs as outlined below (and they can potentially be modified by a user). A CNV is called if z < α, (default: α = 1 × 10−6). We increase the window size in one-base increments and recalculate z to either extend or detect a CNV until a maximum window size wl,max (default = 10,000) is reached. If no CNV has been detected, we move to the next potential breakpoint and repeat our statistical testing. Attempts to extend or detect a CNV will end before reaching wl,max if less than half the bases have abnormal read coverage (≥1.3 or ≤0.70rave,h for diploids). If a CNV was found and wl,max has been reached, we try to extend the CNV by sliding a window of size wl,max and recalculating z. Attempts to extend a CNV continue until thresholds related to read coverage and distance from the CNV end breakpoint have been reached. A flowchart for the sliding window CNV search is provided in Fig. 4.

Figure 4 Flowchart for sliding window CNV search.

For clarity, some conditions for ending a CNV search have been omitted.

Results

Datasets

To test GROM-RD’s performance, we used both simulated (with known SVs) and experimental (with a large number of validated SVs) datasets for a human genome (Table 1). We first compared our approach with two commonly used RD algorithms, CNVnator and RDXplorer, on a simulated dataset. We used RSVSim (Bartenhagen & Dugas, 2013) to simulate 10,000 deletions and duplications ranging from 500 to 10,000 bases using the most recent human reference genome (hg19). RSVSim assumed a beta distribution to create a distribution of CNV sizes based on SVs from the Database of Genomic Variants with lengths between 500 and 10,000 bases, resulting in a decreasing frequency of CNVs with increasing size. Deletions were heterozygous (1 copy number) and duplications ranged from 3 to 10 copy numbers. RSVSim biased SVs to certain types of repeat regions and corresponding mechanisms of formation, such as non-allelic homologous recombination, based on several studies (Chen et al., 2008; Lam et al., 2010; Mills et al., 2011; Ou et al., 2011; Pang et al., 2013). We then used pIRS (Hu et al., 2012) to simulate 100-base Illumina paired-end reads with 500 base inserts and read coverage above ten. pIRS is designed to simulate Illumina base-calling error profiles and GC bias. The simulated reads were mapped to human reference genome hg19 using BWA (Li & Durbin, 2009).

Table 1 Summary of simulated and gold standard datasets.

Dataset	Read length	Insert size	Coverage	Reference	
Simulation	100	500	11x	hg19	
NA12878, low coverage	101	386	5x	hg19	
NA12878, high coverage	101	400	76x	hg18	

We also compared CNVnator, RDXplorer, and GROM-RD on two human datasets (both from NA12878). To better assess algorithm performance with current sequencing technologies (longer reads, lower error rates, etc.), we used the more recent sequence datasets of low coverage NA12878 produced as part of the main project alignments for the 1000 Genomes Project (Abecasis et al., 2012) and high coverage NA12878 produced at the Broad Institute and released to the 1000 Genomes Project (DePristo et al., 2011). Both datasets contain Illumina paired-end reads. We tested algorithm performance using a large set of experimentally validated and high confidence SVs produced during the pilot phase of the 1000 Genome Project and commonly referred to as the “gold standard” (Mills et al., 2011). We will use the term “gold standard” to refer to the above set of validated SVs and the sequence datasets.

Simulation results

CNVnator, RDXplorer, and GROM-RD prediction results for the simulated dataset are shown in Fig. 5. At least 10% reciprocal overlap between a predicted CNV and a simulated CNV was required for a true positive. Default parameters were used for all algorithms, except for the window (bin) size for CNVnator. We estimated the optimal window size for CNVnator (230 bases) by curve fitting the window size and read coverage combinations (resulting in bin size = 2205x−0.941, where x is the read depth) recommended by the program’s authors (Abyzov et al., 2011). The default window size for RDXplorer and GROM-RD is 100 bases. For GROM-RD, we found a 100 base-window to be suitable for all datasets tested.

Figure 5 Sensitivity and FDR for simulated dataset.

GROM-RD had the highest sensitivity and lowest FDR for duplications. GROM-RD’s sensitivity was lower than RDXplorer’s sensitivity for deletions, but GROM-RD had a much lower FDR. Ten thousand deletions and duplications were simulated from human reference hg19 using RSVSim. CNVs were biased to repeat regions. One hundred-base paired-end Illumina reads with 500 base inserts were simulated at 11x coverage using pIRS and mapped to hg19 using BWA.

For the simulated dataset, GROM-RD had the highest sensitivity and lowest false discovery rate (FDR, or the proportion of predictions that were false positives) for duplications. For deletions, our method also had the lowest FDR and second-best sensitivity after RDXplorer, which showed a very high FDR (0.75) when compared to GROM-RD (0.02). When the FDR is very high, it may be more informative to consider the false positive counts. RDXplorer had 13,457 false positives compared to only 61 false positives for GROM-RD. All methods had lower sensitivity and a higher FDR for deletions than duplications, which may be due to the fact that 3 to 10 copy number changes for duplications should be easier to detect than halved RD deletions.

Gold standard results

Prediction results for the gold standard datasets are shown in Table 2. True positives indicate at least 10 or 50% reciprocal overlap between a predicted CNV and the gold standard. CNV predictions not overlapping the gold standard were labeled “Other.” Default parameters were used for all algorithms, except for the window size for CNVnator. Using the previously described curve fitting for CNVnator, we estimated 450 and 100 base windows for the low and high coverage (NA12878) datasets, respectively.

Table 2 CNV prediction results for gold standard datasets.

Results indicate 10%/50% reciprocal overlap between predicted CNV and gold standard. CNV predictions not meeting overlap criteria were classified as “Other.”

	Deletion	Duplication	
Algorithm	Sensitivity	True Positives	Other	Sensitivity	True Positives	Other	
NA12878 (low coverage)	
CNVnator	0.21 / 0.16	102 / 78	548 / 573	0.14 / 0.08	28 / 17	206 / 218	
RDXplorer	0.07 / 0.05	37 / 23	218 / 234	0.03 / 0.01	7 / 3	349 / 355	
GROM-RD	0.44 / 0.37	217 / 181	863 / 901	0.15 / 0.11	31 / 22	313 / 322	
NA12878 (high coverage)	
CNVnator	0.79 / 0.68	391 / 341	27597 / 27653	0.15 / 0.10	34 / 23	975 / 989	
RDXplorer	0.23 / 0.18	117 / 92	1650 / 1679	0.10 / 0.05	22 / 12	794 / 806	
GROM-RD	0.71 / 0.61	352 / 303	5395 / 5438	0.20 / 0.15	45 / 34	1464 / 1472	

Again, GROM-RD had the highest sensitivity for deletions and duplications in the low coverage dataset and duplications in the high coverage dataset. However, CNVnator found 39 more true deletions than GROM-RD in the high coverage dataset with 10% reciprocal overlap or 38 with 50% overlap.

In Table 3, we compared algorithm performance with CNV size (500–10,000 and >10,000 bases) for the gold standard datasets. True positives indicate 10% reciprocal overlap. GROM-RD had the highest sensitivity for all comparisons except for short (500–10k) high coverage NA12878 CNVs. The paucity of supporting evidence makes detecting deletions in low coverage datasets difficult for any method (Fig. 6). However, GROM-RD excelled at detecting deletions in the low coverage dataset, correctly calling more than twice and five times as many deletions as CNVnator and RDXplorer, respectively. Regarding the contribution of individual steps of our pipeline, we note that implementation of the dinucleotide repeat bias adjustment reduced GROM-RD’s deletion predictions in low and high coverage NA12878 by 4 and 48%, respectively, while losing only one true positive prediction. Using quantile normalization for GC bias improved deletion and duplication sensitivity by 768 and 933%, respectively, for low coverage NA12878 and 15 and 73% for high coverage NA12878. Additionally, when employing the two-pipeline approach for excessive coverage masking, deletion and duplication sensitivity increased 6 and 7%, respectively, for the low coverage gold standard dataset and 4 and 15% for the high coverage gold standard dataset.

Figure 6 Example of deletion in chromosome 1 of NA12878 (detected only by GROM-RD in the low coverage dataset).

Histogram at the top reflects read coverage across the region. Grey pointed rectangles connected by lines represent paired reads. Gold standard validation (108402984–108405403) and GROM-RD’s prediction (108402966–108405569) are represented by the black and blue double-arrowed lines, respectively. CNVnator and RDXplorer did not predict the deletion. We note that deletions in low coverage datasets are difficult for any method to detect as evidenced by only one discordant read pair (red pointed rectangles connected by red line) supporting the deletion making detection unlikely by a PR method. Example is shown using human reference hg19 in IGV viewer (Robinson et al., 2011).

Table 3 Comparison of algorithm performance for different CNV sizes.

Results shown for short (500–10,000 bases) / long (>10,000 bases) CNVs. True positives indicate 10% reciprocal overlap. CNV predictions not meeting overlap criteria were classified as “Other.”

	Deletion	Duplication	
Algorithm	Sensitivity	True positives	Other	Sensitivity	True Positives	Other	
	NA12878 (low coverage)	
CNVnator	0.11 / 0.72	47 / 55	202 / 346	0.03 / 0.27	3 / 25	20 / 186	
RDXplorer	0.03 / 0.34	11 / 26	62 / 156	0.03 / 0.04	3 / 4	217 / 132	
GROM-RD	0.37 / 0.84	153 / 64	740 / 123	0.05 / 0.27	6 / 25	86 / 227	
	NA12878 (high coverage)	
CNVnator	0.78 / 0.81	328 / 63	27132 / 465	0.09 / 0.23	12 / 22	618 / 357	
RDXplorer	0.16 / 0.62	69 / 48	1418 / 232	0.05 / 0.15	7 / 15	595 / 199	
GROM-RD	0.68 / 0.83	287 / 65	5413 / 156	0.15 / 0.26	20 / 25	1216 / 252	

Breakpoint accuracy

Breakpoint accuracy is one of the traditional weaknesses of the RD methods and improvements in this area can help in narrowing down CNV borders and facilitate subsequent validation experiments. CNVnator, RDXplorer, and GROM-RD breakpoint accuracy on the simulated and NA12878 gold standard datasets is summarized in Table 4. GROM-RD had the lowest deletion and duplication breakpoint error for all datasets, except duplications for low coverage NA12878 where RDXplorer had lower breakpoint error (11823 bases) compared to GROM-RD (22555). We note that RDXplorer had only seven true positive duplication calls for low coverage NA12878, limiting the reliability of the breakpoint error estimation. We observed larger breakpoint error for the NA12878 gold standard datasets relative to the simulation dataset. This was partly due to the simulation study not having large CNVs (>10k) which had larger breakpoint error compared to short (500–10k) CNVs in the gold standard datasets. Additionally, breakpoints have complexities (such as microhomology of sequence around breakpoints, repeat sequences, etc.) that are not well understood and simulated.

Table 4 Mean breakpoint error for simulated and gold standard datasets.

Lowest error for each measurement is bolded. GROM-RD had the lowest deletion (Del) and duplication (Dup) breakpoint error for all datasets.

	Simulation	NA12878 (low coverage)	NA12878 (high coverage)	
Algorithm	Del	Dup	Del	Dup	Del	Dup	
CNVnator	278	303	8,486	47,057	2,846	23,729	
RDXplorer	270	147	23,587	11,823	8,454	27,122	
GROM-RD	128	91	4,687	22,555	2,025	13,536	

Algorithm metrics

Run times for the algorithms on the gold standard datasets are provided in Table 5. We tested all three programs on a single CPU (Intel Xeon E31270, 3.4 GHz) on a Linux workstation with 16 GB RAM memory. Standard BAM files were used as input. In contrast to other tools, GROM-RD’s run time is relatively insensitive to read coverage with a 15-fold increase in coverage resulting in only a 33% increase in run time. GROM-RD is written in C, uses standard BAM files as input, is able to utilize paired or single reads, and is available at http://grigoriev.rutgers.edu/software/.

Table 5 Run times (in minutes) on gold standard datasets.

Algorithm	Low coverage (NA12878)	High coverage (NA12878)	
CNVnator	47	206	
RDXplorer	371	4378*	
GROM-RD	112	149	
Notes.

* RDXplorer outputs very large files, low I/O throughput may have affected the run time for this dataset significantly.

Discussion

We developed a novel RD approach for detecting CNVs in NGS data. Many RD algorithms, such as CNVnator and RDXplorer, correct GC bias by binning genome regions based on GC content and normalizing the read depth mean of each bin to the global average. However, read depth variance tends to vary with GC content after normalizing the means (Fig. 2). GROM-RD normalizes variance by using a quantile normalization approach to convert read depth to standard deviation units. As a result, our method produces fewer false positives overall. GROM-RD, CNVnator, and RDXplorer were tested on a simulated and two gold standard datasets. GROM-RD performed well on the simulated data having the highest sensitivity and lowest FDR. Although RDXplorer had a somewhat higher sensitivity for deletions compared to GROM-RD, it came at the expense of extreme overprediction: RDXplorer had a very high FDR resulting in 13,457 false positives compared to only 61 false positives for GROM-RD. GROM-RD had the highest sensitivity for deletions and duplications on the low coverage gold standard dataset and for duplications on the high coverage gold standard dataset. For deletions in the high coverage dataset, GROM-RD had comparable sensitivity (0.71) to CNVnator (0.79). GROM-RD excelled at detecting deletions in the low coverage NA12878 dataset, correctly calling more than twice and five times as many deletions as CNVnator and RDXplorer, respectively. When comparing performance by CNV size, GROM-RD had the highest sensitivity for all comparisons except for short (500–10k) high coverage NA12878 CNVs, where GROM-RD had comparable sensitivity (0.68) to CNVnator (0.78). GROM-RD’s dinucleotide repeat bias normalization reduced GROM-RD’s deletion predictions by 4 and 48% on the low and high coverage datasets, respectively, while losing only one true positive, suggesting an improvement in specificity. As expected, duplication predictions were not affected by dinucleotide repeat bias normalization. Using quantile normalization for GC bias normalization improved deletion and duplication sensitivity by 768 and 933%, respectively, for low coverage and 15 and 73%, respectively, for high coverage NA12878. Compared to one pipeline with no excessive coverage masking, our two pipeline approach with excessive coverage masking increased deletion and duplication sensitivity 6 and 7%, respectively, for the low coverage gold standard dataset and 4 and 15% for the high coverage gold standard dataset.

Often RD algorithms analyze read depth in non-overlapping windows with a fixed size. A read is placed in a window if the read’s center (CNVnator) or start (RDXplorer) occurs in the window. Fixed-size, non-overlapping windows result in low breakpoint resolution. GROM-RD utilizes sliding windows with sizes varying in one-base increments to improve breakpoint accuracy. For all datasets except duplications for low coverage NA12878, GROM-RD had the lowest deletion and duplication breakpoint error, thus improving this common weakness of RD methods.

RD algorithms are complementary to and have some advantages compared to other CNV detection methods. For instance, RD algorithms may be able to detect CNVs with rough breakpoints and duplications with few uniquely mapped reads that paired- and split-read methods may have difficulty detecting. We observed a number of such cases for validated CNVs in the low coverage NA12878 dataset, with just one discordant read pair spanning a deletion (Fig. 6) or even with no support from discordant paired reads at all. However, RD methods frequently have low breakpoint resolution. Our results suggested that GROM-RD was able to improve RD sensitivity, specificity, and breakpoint accuracy compared to CNVnator and RDXplorer, the two most frequently used RD algorithms. Additionally, GROM-RD had a short run time that was relatively insensitive to read coverage indicating excellent scalability of the method for different datasets.

We thank Kevin Abbey and Sulbha Choudhari of Rutgers University for excellent technical help and advice throughout the development and testing process.

Additional Information and Declarations

Competing Interests

Author Contributions

The authors declare there are no competing interests.

Sean D. Smith performed the experiments, analyzed the data, contributed reagents/materials/analysis tools, wrote the paper, prepared figures and/or tables, reviewed drafts of the paper.

Joseph K. Kawash analyzed the data, reviewed drafts of the paper.

Andrey Grigoriev conceived and designed the experiments, analyzed the data, wrote the paper, reviewed drafts of the paper.

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
