# Peer review of "GROM-RD: resolving genomic biases to improve read depth detection of copy number variants"

_PeerJ, doi:10.7717/peerj.836_

## Round 0.1 · original submission · Major Revisions

We had two reviewers go over the manuscript and the advice ranged from minor to major revisions may be needed; however, each reviewer did have a positive response for the method tested. The minor review outcome also came away with some additional proof being desired. The general overall opinion was that more information would be needed to help explain why the method appeared to work over other software tools available. Perhaps identification of some parameters which would highlight where the software improves the results, or some monitoring of progress in steps used by other protocols would help prove your conclusions. You did try to benchmark GROM-RD against CNVnator and RDXplorer, but I think some parameter or comparison of each method might shed some light on the observed performance boost. Based on these assessments I would place this manuscript in the major revision category, but if some documentation might be provided to clearly highlight the differences of this method over others, and to perhaps hone-in on a parameter that may be a main factor affecting the results, then a case to move this manuscript forward would be more favorable. I would suggest going over the reviewers notes and trying to best address their queries. I think many would like to use the tools and I think this may also help build a discussion on what is really needed in resolving copy number variants.

Thank you for your contribution and I would hope for a short turn-around in addressing the review notes.

Reviewer 1 ·

Basic reporting

The authors have reported a new method based on read-depth signature for CNV discovery. They claim that after GC-bais correction done by all the RD based methods, their is a difference of variance for different regions based on the GC.

Experimental design

No Comments

Validity of the findings

The simulation and real data results are convincing. However, I am not able to understand why their method has such an improvement over the competitions considering that I could not see a major difference. It would be ideal if they show in one example how/why their method is outperforming others for instance CNVnator.

Reviewer 2 ·

Basic reporting

The structure of the manuscript and language are fine.

Experimental design

Authors try to address important questions, however many technical details are omitted which make difficult to evaluate the results. Namely:
* it is not clear how CNVs were simulated. Was CNV size chosen uniformly between 500 and 10,000 bps? How was location of CNV in genome chose?
* Sliding/multiple window CNV search is not clear at all. Perhaps, authors can provide a diagram clarifying it.
* It is not clear where gold standard comes from. Is it the same standard as the one utilized by Mills et al.? If so, were performance of RDXplorer and CNVnator in authors' hands the same as described by Mills et al.?

Validity of the findings

The major claim of the paper that the described methods GROM-RD improves compare to previous methods. While authors provide some good arguments to support this claim, they come short in providing a more detailed.
* It is not clear whether the improvements they claim is universal for CNVs of all size length and/or genomic locations.
* Could authors quantify what was the contribution of each improvement towards better performance: GC correction, di-nucleotide bias correction, multi-window calling.
* I'm not certain that 10% reciprocal overlap is a good measure. How would results change if one utilizes a different fraction of overlap, e.g., 50%? Perhaps, 2D plot of reciprocal overlap between calls by different methods will be informative.
* It is hard to believe that CNVnator called 27,597 deletions (Table 2) for NA12878. Particularly, because NA12878 is sequenced by the 1000 Genomes Project and CNVnator was applied to that genome in the original publication. In fact, after quick checking of the original CNVnator publication, it seems that for the same genome few thousand of call were reported. Can the authors comment on the difference?
* Breakpoint accuracy was drastically different on simulated and real data for all methods. Could author comment on this. I'm afraid that the simulation used in this study may not adequately reflect real CNVs.

Additional comments

This is an interesting study that has many weaknesses and shortcomings.

---

## Round 0.2 · accepted · Accept

This edited version of the original manuscript read very cleanly and addressed many of the concerns presented by the reviewers. The manuscript now presents case by case data evaluations for readers to compare and points to the current benchmarks available. Highlighting the strengths of current tools available will provide readers a starting point to familiarize themselves which what GROM-RD can provide.

Your supplemental data is made available as a single GZIPed file from your personal laboratory website; it may be best to have it maintained at a more stable data repository resource. PeerJ has limited space for supplemental data, or another public data commons resource may be available. Some of your archived material draws from the samtools resource from GitHub; perhaps you might point to that resource, and provide your original code separately. That may help reduce the footprint of the archived program source files.

There were only a couple suggested edits:

line 170: “dinucleote”
[mispelled? And maybe reword the sentence to avoid repetition of the string “dinucleotide repeat bias”].

Figure 5: Y-axis needs to be labelled.

With other related tools available, I hope that this will add to discussion on what is really needed in resolving copy number variants. Congratulations. Also, since your manuscript submission PeerJ has added a new journal area, PeerJ Computer Science; however, I feel this work will well serve the bioinformatics readership. Thank you for your contribution.